# A Review of Biomarkers of Amyotrophic Lateral Sclerosis: A Pathophysiologic Approach

**DOI:** 10.3390/ijms252010900

**Published:** 2024-10-10

**Authors:** Rawiah S. Alshehri, Ahmad R. Abuzinadah, Moafaq S. Alrawaili, Muteb K. Alotaibi, Hadeel A. Alsufyani, Rajaa M. Alshanketi, Aysha A. AlShareef

**Affiliations:** 1Department of Physiology, Faculty of Medicine, King Abdulaziz University, Jeddah 22252, Saudi Arabia; rsjalshehri@kau.edu.sa (R.S.A.); haalsufyani@kau.edu.sa (H.A.A.); 2Department of Neurology, Faculty of Medicine, King Abdulaziz University, Jeddah 22252, Saudi Arabia; alruily@kau.edu.sa (M.S.A.); aaalshareef@kau.edu.sa (A.A.A.); 3Neuromuscular Medicine Unit, King Abdulaziz University Hospital, King Abdulaziz University, Jeddah 22252, Saudi Arabia; 4Neurology Department, Prince Sultan Military Medical City, Riyadh 12233, Saudi Arabia; mk-alotaibi@psmmc.med.sa; 5Internal Medicine Department, King Abdulaziz University Hospital, King Abdulaziz University, Jeddah 22252, Saudi Arabia; ralshanketi@kau.edu.sa

**Keywords:** amyotrophic lateral sclerosis, CSF, blood, biomarker, pathophysiology

## Abstract

Amyotrophic lateral sclerosis (ALS) is a neurodegenerative disease characterized by progressive degeneration of upper and lower motor neurons. The heterogeneous nature of ALS at the clinical, genetic, and pathological levels makes it challenging to develop diagnostic and prognostic tools that fit all disease phenotypes. Limitations associated with the functional scales and the qualitative nature of mainstay electrophysiological testing prompt the investigation of more objective quantitative assessment. Biofluid biomarkers have the potential to fill that gap by providing evidence of a disease process potentially early in the disease, its progression, and its response to therapy. In contrast to other neurodegenerative diseases, no biomarker has yet been validated in clinical use for ALS. Several fluid biomarkers have been investigated in clinical studies in ALS. Biofluid biomarkers reflect the different pathophysiological processes, from protein aggregation to muscle denervation. This review takes a pathophysiologic approach to summarizing the findings of clinical studies utilizing quantitative biofluid biomarkers in ALS, discusses the utility and shortcomings of each biomarker, and highlights the superiority of neurofilaments as biomarkers of neurodegeneration over other candidate biomarkers.

## 1. Introduction

Amyotrophic lateral sclerosis (ALS) is a fatal neurodegenerative disease characterized by progressive degeneration of the anterior horn cells of the spinal cord, motor brainstem nuclei innervating bulbar muscles, and degeneration of corticospinal motor neurons [1]. This pathognomonic loss of both upper and lower motor neurons results in progressive paralysis of spinal and bulbar innervated skeletal muscles and inevitable death within approximately three to five years of symptoms onset. Degeneration starts focally in the form of a spinal onset in two-thirds of patients and a bulbar onset in one-third of patients before contiguous dissemination to other areas [2]. Concomitant degeneration of the frontotemporal lobes is reported to occur in at least 25% of patients [3]. ALS primarily affects patients in late adulthood, with an average age of onset of 65 years [4]. Most ALS cases (90%) are sporadic (sALS) with no family history of ALS, while the remaining minority (10%) are familial (fALS) [1]. So far, over 40 genes have been linked to 11% of sporadic cases and nearly 70% of familial cases [5,6]. More genetic mutations are likely to be uncovered as research progresses. Diagnosis of ALS entails the exclusion of mimic conditions in the presence of clinical evidence of spatiotemporal progression of upper motor neuron (UMN) involvement combined with clinical or electrophysiological evidence of progressive lower motor neuron (LMN) involvement [7]. Despite the disease’s fast progression, diagnostic delay ranges around 12 months, which is partly attributable to the disease’s heterogeneity and a lack of a specific biomarker [1,8]. Diagnostic delay of an average of one year for a disease that has a survival of three to five years is not only burdensome but could potentially narrow a possible therapeutic window [3]. Indeed, confirmation of ALS diagnosis can cost a patient up to USD 20,000, highlighting the financial burden and the psychological burden of the journey to diagnosis [9]. New diagnostic criteria have been recently developed to advance diagnostic sensitivity by essentially including progressive muscular atrophy as a variant of ALS. Despite the Gold Coast criteria having the highest sensitivity of the established criteria, they have a false negative rate of nearly 10% [10]. Furthermore, about 10% of ALS-diagnosed cases are later revealed to be disease mimics [11], emphasizing the need for highly sensitive and specific diagnostic tools. Limitations associated with the functional scales and the qualitative nature of mainstay electrophysiological testing prompt the investigation of non-invasive and more objective prognostic tools [12]. Biomarkers have the potential to fill that gap by providing evidence of a disease process potentially early in the disease, its progression, and its response to therapy. Therapeutic options for ALS remain to be limited, as the only globally licensed drug (riluzole) offers a modest improvement in survival of a mere few months [13]. Advancement in understanding of the variable pathophysiological mechanisms facilitated the development of numerous experimental therapies [14,15,16]. However, the overwhelmingly unsuccessful translation of several previous experimental therapies in clinical trials [17] highlights the intricateness of the disease and the need for more effective pharmacodynamic biomarkers [18].

In contrast to other neurodegenerative diseases (NDs), such as multiple sclerosis (MS) and Alzheimer’s disease (AD), no biomarker has yet been validated in clinical use for ALS. Fluid biomarkers offer accessible quantitative measurements that can facilitate diagnosis, monitor progression, help identify therapeutic targets, and assess response to therapy. They are patient-convenient, reproducible, and relatively cost-effective [19]. As the incidence of ALS is expected to increase exponentially by 2040 [12], the need for established disease biomarkers grows even greater.

Several fluid biomarkers have been investigated in clinical studies of ALS reflecting the different pathophysiologic processes, from protein aggregation to muscle denervation. This review takes a pathophysiologic approach to summarizing the findings of clinical studies utilizing quantitative biofluid biomarkers in ALS. Structure-based biomarkers [20,21] and function-based biomarkers [22] are outside the scope of this review.

## 2. Pathophysiology of Amyotrophic Lateral Sclerosis

Although our understanding of the pathophysiology of ALS remains incomplete, the identification of several pathogenic mutations has revealed various mechanisms. The most studied ALS-linked genes are chromosome 9 open reading frame (*C9orf72)*, Cu^2+^/Zn^2+^ superoxide dismutase 1 (*SOD1*), TAR DNA binding protein (*TARDBP*), and fused in sarcoma (*FUS*), which collectively account for nearly 48% and 5% of familial and sporadic cases, respectively [23]. Mutations in *TARDBP* and *FUS* lead to aggregation of the DNA and RNA binding proteins TARDBP 43-kDa (TDP-43) and FUS, respectively [7]. Mutations in the *C9ORF72* gene lead to dipeptide aggregates, intranuclear RNA deposits, and, interestingly, TDP-43 aggregates [3]. Mutations in the *SOD1* gene result in aggregation of the mutant SOD1, mitochondrial dysfunction, and consequently perpetuation of oxidative stress. Other less common mutations, such as tubulin alpha 4a (*TUBA4A*) and profilin 1 (*PFN1*), have been linked to disruption of axonal transport through cytoskeletal and tubulin defects [7].

It used to be believed that the core pathology of ALS is aberrant aggregation of mislocalized or misfolded proteins in motor neurons [24]. This is supported by the close association of neuronal loss with the burden of the proteinaceous inclusions in postmortem tissues [25]. Protein aggregation follows protein misfolding or mislocalization due to mutations, oxidation, starvation, or cross-seeding. Protein aggregation interferes with cellular functions such as axonal transport, mitochondrial respiration, and stress response [26]. Prion-like spreading of proteinaceous inclusions is seen in both familial and sporadic ALS [2]. In 97% of ALS cases, there are cytoplasmic aggregates of mislocalized, ubiquitinated phosphorylated TDP-43, which normally works as a transcription factor in the nucleus [7]. TDP-43 normally shuttles between the nucleus and the cytoplasm and has a prion-like domain, allowing it to accumulate into stress granules when needed. Stress induces the exit of the protein from the nucleus into stress granules, from which it can shuttle back to the nucleus after stress resolution. In ALS, it is irreversibly mislocalized to the cytoplasm either due to mutations or likely accumulated unresolved stress. The mislocalization of TDP-43 is believed to instigate sequelae of pathological alternations, including failure of mRNA splicing and subsequent depletion of certain proteins or, indeed, the development of proteins containing aberrant sequences [26]. Additionally, aggregation of TDP-43 prevents the clearance of damaged proteins, causing impaired autophagy [7]. FUS is another RNA-binding protein with a prion-like domain and a propensity to accumulate in stress granules. TDP-43 and FUS share several pathogenic features as their aggregates not only sequester proteins involved in RNA binding but can also recruit multiple proteins to aggregate through cross-seeding. Thus, they are aggregation-inducing proteins making way for the disease to propagate through contiguous cells [26]. SOD1 is another protein with prion-like properties when misfolded. SOD1 aggregation disrupts mitochondrial respiration, axonal transport, and normal proteostasis through mechanisms that remain to be elucidated [26]. Although the aforementioned toxic aggregates (TDP43, SOD-1, and FUS) share pathophysiological mechanisms, they do not coexist [26]. Conversely, TDP43-positive inclusions coexist with dipeptide repeat (DPR) inclusions in ALS caused by mutations in C9ORF72. Mutations in *C9ORF72* are nucleotide repeat expansion mutations, which account for 40% of familial ALS cases. Identification of this mutation has informed a less protein-centric approach to ALS pathogenesis, marking aberrant RNA processing as a key mechanism for ALS. This is further supported by *TDP43* and *FUS*’s role in RNA processing [26].

ALS is now recognized as a non-cell autonomous disease that involves non-neuronal cells as facilitators of neuronal death [26]. Postmortem studies show low expression of the excitatory amino acid transporter 2 (EAAT2) in the brain and spinal cord of ALS patients. This is likely due to the deregulated processing of the EAAT2 mRNA transcript, which results in high glutamate concentration. High glutamate concentration activates N-methyl D-aspartate (NMDA) and alpha-amino-3-hydroxy-5-methyl-4-isoxazolepropionic acid (AMPA) receptors, resulting in excessive calcium influx and subsequent calcium overload. Calcium overload, which is primarily mediated by the glutamate receptor 2 (GluR2) subunit of AMPA receptors, overwhelms the mitochondria, resulting in oxidative stress [27]. Additionally, calcium overload over-activates glial cells, which exacerbates free radical production, thereby perpetuating oxidative stress [28]. Furthermore, calcium overload activates certain enzymes such as endonucleases, proteases, phosphatases, and phospholipases, which propagates excitotoxic cell death [27,29]. Neurons are generally susceptible to glutamate-induced excitotoxicity due to the inability to regenerate and permeability of activated glutamate receptors. This susceptibility of neurons can be exacerbated by existing mitochondrial dysfunction and oxidative stress [29]. Motor neurons are especially susceptible to excitotoxicity due to limited stress response, limited calcium buffering ability, and high expression of calcium-permeable AMPA receptors [26].

Disruption of the blood–brain barrier (BBB) and the blood–spinal cord barrier (BSCB) is believed to occur in the early stage of the disease. Barrier disruption and glial activation allow for macrophage infiltration into the central nervous system (CNS). Neuroinflammation accelerates disease progression via an increase in cytotoxic immune cells as well as other mechanisms described in previous reviews [23,27]. Unlike other NDs, ALS progresses much more rapidly and with more heterogeneity between patients [30]. Whether the contiguous spread of motor neuronal death occurs via anterograde degeneration or retrograde degeneration is debatable. It is likely, however, that the two processes occur concurrently [31]. The result of both processes is the disassembly of the neuromuscular junction and the denervation of skeletal muscles [32].

It is evident that several overlapping mechanisms are implicated in the pathogenesis of ALS, as illustrated in Figure 1. Motor neuron degeneration is believed to be the result of multiple hits, comprising protein misfolding, aberrant protein aggregation, neuroinflammation, oxidative stress, mitochondrial dysfunction, impaired RNA processing, glutamate excitotoxicity, and disturbed axonal transport [3,26]. These mechanisms appear to be interrelated and likely downstream of unknown primary insults triggered by the interaction of genetic, epigenetic, and environmental factors [33].

## 3. Biomarkers in Amyotrophic Lateral Sclerosis

An ideal biomarker for an ND would need to have the following characteristics. The biomarker should be stable in the fluid from which it is measured. The biomarker should be sensitive to ongoing injury and preferably specific to the underlying pathology [34]. Its release into the surrounding cerebrospinal fluid (CSF) or through the BBB into the plasma should be passive and, therefore, reflective of the extent of pathology [35,36,37,38]. The biomarker should be easily isolated and analyzed, and its collection should be minimally invasive for patients [39]. CSF has the advantage of being in direct contact with the CNS, but its use is limited by the invasiveness of lumbar puncture. On the other hand, blood sampling is less invasive, but its analysis is ultimately more complex as it contains a variety of proteins [40].

Such biomarkers can be diagnostic, prognostic, predictive, or pharmacodynamic. Diagnostic biomarkers can distinguish those affected by the disease from those that are not. Diagnostic biomarkers can expedite diagnosis, therapeutic administration, and enrollment in early disease clinical trials [41]. Predictive biomarkers determine which patient will likely experience an outcome in response to a clinical intervention. Validation of a predictive biomarker therefore requires a controlled trial including patients with and without the biomarker [42]. Pharmacodynamic biomarkers can illustrate proof of targeting and downstream effector activity of candidate therapies to determine whether the therapeutic target has been achieved. On the other hand, prognostic biomarkers can predict an outcome, such as the rate of disease progression and severity, irrespective of treatment, thereby helping manage patients’ expectations and allocating appropriate healthcare plans [41,42].

Given ALS’s complex and intertwined pathophysiology, it is ambitious to expect one biomarker to tick all the boxes of an ideal biomarker. This article reviews evidence investigating the diagnostic, prognostic, predictive, and pharmacodynamic potential of quantitative biofluid ALS biomarkers.

### 3.1. Biomarkers Related to Proteinopathy

ALS is a proteinopathy in which the deposition of insoluble intracellular proteinaceous inclusions along degenerating neurons is a core feature [43]. These proteinaceous inclusions (TDP-43, FUS, and SOD1) originate from mislocalized or misfolded proteins that then disseminate along adjacent pools of motor neurons in a self-seeding way similar to that of prion proteins [20,44]. In ALS, spatiotemporal neurodegeneration dissemination correlates with the extent of proteinopathy [25]. Although mutations in *TARDBP*, the gene that encodes TDP43, are an uncommon cause of ALS, hyperphosphorylated and ubiquitinated cytoplasmic aggregates of TDP-43 are disseminated across the spinal cord and motor cortex in the overwhelming majority of patients [26].

TDP-43 is a DNA- and RNA-binding protein of 414 amino acids predominantly expressed in the CNS. It is a transcription factor involved in RNA processing and post-transcriptional regulation [43,45]. Its C-terminal glycine-rich region, which harbors the bulk of mutations, is involved in stress response, protein–protein interactions, and the formation of intracellular RNA granules [24]. TDP43 proteinopathy is a unifying pathology in most ALS cases where TDP-43 mislocalizes partially or entirely from the nucleus to the cytoplasm. The mechanism behind the mislocalization of TDP-43, which is essential to instigating the pathological alternations of sALS, is not fully understood [13]. Some researchers hypothesize that mislocalization of TDP-43 leads to nuclear depletion of the protein and loss of its function [46,47]. In contrast, others argue that TDP-43 proteinopathy is attributed to gain of function via the toxicity of the aggregates in the cytoplasm [48]. In either hypothesis, TDP-43 mislocalization initiates a cascade of neurodegenerative changes, as shown by a study where pathological TDP-43 injection into experimental cerebral organoids replicated ALS pathophysiology [49].

TDP-43 proteinopathy has been associated with altered stress granule dynamics, sequestration of sequestosome-1, mitochondrial dysfunction, loss of axonal transport, autophagy deregulation, and impaired endocytosis [43,45]. Accumulation of mislocalized TDP-43 not only overwhelms physiological clearance systems (proteasomal, endosomal, and autophagic) [7,50] but seems to interfere with the functioning of proteins involved in protein degradation, such as valosin-containing protein (VCP) [26]. Additionally, TDP43 aggregates have an indirect toxic proinflammatory effect via the activation of microglia through NF-κB and AP-1 pathways and the intracellular inflammasome [51].

Several forms of TDP-43 are observed in ALS, including hyperphosphorylated, C-terminus fragmented, and acetylated [52,53,54,55]. This complicates the feasibility of TDP-43 as a therapeutic target [56] and its use as a biomarker. Still, TDP-43 has been detectable in the cerebrospinal fluid (CSF) using enzyme-linked immunosorbent assay (ELISA) [57,58,59], single-molecule array (Simoa) [60,61] and quantitative mass spectrometry [62]. Additionally, TDP-43 has been detectable in plasma using ELISA [59,63], Simoa [60], and quantitative mass spectrometry [62], as well as in platelets using ELISA [64]. CSF TDP-43 levels are significantly more elevated in ALS patients compared to neurological disease controls (NDCs) in two cohorts using ELISA [57,58]. Interestingly, CSF levels were negatively associated with survival in the latter study, hypothetically due to insoluble deposits on tissues leading to lower extracellular levels. However, no neuropathological study was performed to support this hypothesis [57]. In a recent cohort, plasma TDP-43 levels were significantly higher in ALS patients than in healthy controls (HCs), with a sensitivity and specificity of over 90% and an AUC value of 0.924. Additionally, plasma TDP-43 levels correlated with the ALS Functional Rating Scale (ALSFRS-R) and time to generalization (TTG). In the same study, CSF TDP-43 levels correlated with plasma levels but had less diagnostic potential with an AUC of 0.588 and had lower concentrations than in plasma. The researchers argued that lower TDP-43 levels in the CSF compared to plasma are attributed to protease in the CSF, which would affect the protein stability [59]. Indeed, it is estimated that TDP-43 has a 200 times higher concentration in the plasma than in the CSF [62]. This finding was replicated in another study where CSF TDP-43 levels correlated with plasma levels but were significantly lower. However, in this cohort using Simoa, CSF-TDP43 had a higher AUC than in plasma, though neither correlated with survival [60]. Conversely, plasma TDP-43 levels showed a negative correlation with the split hand index and vital capacity percentage in another study using Simoa [61]. It is suggested that the biomarker potential of TDP-43 is constrained by antibody-based assays as the antibodies bind the pathological and physiological form of the protein [62,65,66]. However, studies using ELISA have yielded promising results [57,58,59], and it is likely that as the detection techniques are further improved, more conclusive evidence will come to light.

FUS is another RNA-binding protein with a prion-like pro-aggregating domain implicated in ALS pathogenesis. It is a nuclear 526 amino-acid protein primarily involved in RNA processing [67,68,69]. It has several functional domains, but its C-terminal nuclear localization signal is where the bulk of mutations occur [70]. Mutations in the FUS gene account for 3% of fALS cases and 0.3% of sALS [20,71]. Mutations in the *FUS* gene interrupt transportin-mediated nuclear import, resulting in the mislocalization of the FUS protein in the cytoplasm and subsequent sequestration into stress granules [26]. The exact role of mislocalized FUS in neurodegeneration is unclear, but research suggests that the sequestration of FUS disrupts the nuclear localizing signal and RNA processing. Still, a second hit is needed for the mislocalized FUS to aggregate and disrupt RNA processing. For example, in an experimental study, mislocalization combined with oxidative stress contributed to the sequestration and aggregation of FUS [72]. Research shows that microvesicles derived from ALS patients with FUS pathology have significantly higher FUS levels than controls [20]. However, the protein’s potential as a fluid biomarker is not supported by any data, likely due to its rare occurrence.

Another protein implicated in ALS pathogenesis involves SOD1, a free radical scavenging enzyme essential to counteract oxidative stress [27]. Genetic mutations in *SOD1* account for 20% and 1% of fALS and sALS, respectively [26]. The SOD1 gene codes for the SOD enzyme, which catalyzes the dismutation of superoxide (O2−) to hydrogen peroxide (H2O2) and oxygen (O2) [73]. The toxicity of SOD1 mutations is likely related to oxidative stress, mitochondrial dysfunction, disruption of the proteasomal pathway, and autophagy [26]. Protein aggregation of misfolded SOD1 is self-seeding in that newly formed aggregates cause subsequent misfolding of native SOD1[74]. The significance of SOD1 aggregation is highlighted by its correlation with the rate of disease progression [75]. Lowering levels of misfolded SOD1 is a therapeutic target that shows promise in preclinical studies. Tofersen is an FDA-approved *SOD1* silencing agent aimed to inhibit the synthesis of SOD1 protein by inducing RNase [76]. In experimental *SOD1* disease models, targeting *SOD1* with antisense oligonucleotides (ASO) decreased *SOD1* mRNA and protein levels and slowed disease progression [77,78,79].

Although primarily intracellular, SOD1 can be secreted [80] and has been detected in the CSF of ALS patients by ELISA [78,81] and liquid chromatography-mass spectrometry (LC-MS) [82]. Using LC-MS, researchers were able to detect significantly elevated levels of a specific SOD1 peptide in the CSF of 13 presymptomatic and fourteen symptomatic *SOD1* mutation carriers compared to disease controls comprising thirty patients with NDs and 29 HCs. This assay had a sensitivity of 92.6%, a specificity of 80%, and an AUC of 0.96 for *SOD1* mutations regardless of symptoms [82]. Additionally, SOD1’s potential as a pharmacodynamic biomarker was recently highlighted in the VALOR study as a longitudinally stable protein. In the study, intrathecal administration of tofersen resulted in a significant reduction of CSF SOD1 levels in the treatment group (*n* = 72) compared to patients in the placebo group (*n* = 36) at 28 weeks, although this was not met with significant clinical improvement. Interestingly, in the subgroup analysis, slow progressors in the placebo group had a reduction of CSF SOD1 levels by 19%, while fast progressors in the placebo group had an increase of CSF levels by 16% [83]. Larger cohorts using ELISA failed to establish a comparable diagnostic potential of CSF SOD1. While CSF SOD1 levels were higher in ALS patients compared to HCs, they were similar to that of the NDCs comprising AD, MS, and peripheral neuropathy in one cohort [78] and an undefined disease control group in another [81]. CSF SOD1 levels did not correlate with the disease characteristics in either study but were stable upon longitudinal measurement [78,81]. The relevance of SOD1 as a biomarker in other biological fluids is negated by its elevation in several non-neurological disorders [84,85,86].

Pathogenic expansion of a hexanucleotide repeat GGGGCC in *C9ORF72* is the most common genetic cause of ALS, accounting for 40% of fALS [26]. This mutation results in loss of function of the C9ORF72 protein and the generation of non-ATG translation of repeated RNA sequences producing five DPR proteins namely, poly-glycine–alanine (poly-GA), poly-glycine–arginine (poly-GR), poly- glycine–proline (poly-GP), poly-proline–alanine (poly-PA) and poly-proline–arginine (poly-PR) [87]. These DPR proteins aggregate into intraneuronal inclusions alongside TDP43-positive inclusions [26]. The five classes of DPR proteins have been implicated in neurodegeneration through caspase activation and inhibition of membrane-less organelle formation, such as nucleoli and stress granules [88,89]. Arginine-containing DPRs, however, appear to be more toxic (poly-GR and poly-PR), likely due to their role in post-translational modification by methyltransferases, which gives rise to methylarginine DPR [90]. Targeting this mutation through ASO, thereby obliterating repeat RNA and DRP protein production, is being investigated [8]. CSF levels of three classes of DPR proteins have been studied in ALS. Poly–GA and Poly-GR have been detectable using meso-scale discovery (MSM)-based immunoassays [91], while poly-GP has been detectable using MSM [18,92] and Simoa [93]. In all studies, CSF levels of poly-GP, poly-GA, and poly-GR were similar in presymptomatic and symptomatic *C9orf72* repeat expansion carriers and were undetectable in non-carriers [18,91,92,93], but this had no clinical correlation [18,92]. Additionally, poly-GR and poly-GP are candidate pharmacodynamic biomarkers as their levels appear to be stable over time [18,91]. Indeed, treatment with ASO targeting the *C9ORF72* transcript decreased CSF poly-GA, and poly-GR levels by approximately 50% within six weeks in one patient, but this has yet to be reexamined in larger studies [91].

### 3.2. Biomarkers Related to Neurodegeneration

Degeneration of UMNs and LMNs is a pathognomonic feature of ALS [94]. Degeneration of motor neurons leads to scarring along the lateral tracts of the spinal cord and progressive muscle paralysis [26]. Degeneration generally starts focally before disseminating contiguously across UMN and LMN levels [94]. It is unknown whether the disease spreads through anterograde transneuronal degeneration originating in the primary motor cortex or retrograde degeneration starting in the lower motor neurons or both [31]. While neuropathological studies show more abundant neuronal inclusions in LMNs compared to UMNs [94], transcranial magnetic stimulation studies show evidence of cortical involvement preceding the onset of LMN dysfunction [95].

Neuron-specific enolase (NSE) is a glycolytic enolase isozyme primarily expressed in neurons and neuroectodermal cells [96]. It is passively secreted into the CSF in response to any form of neuronal damage [97]. Therefore, CSF levels are reflective of the extent of neuronal injury [98]. In ALS, NSE has been detectable in CSF using electro-chemiluminescent immunoassay (ECLIA). NSE can differentiate ALS from cervical spondylotic myelopathy and NDs with an AUC of 0.86 [98]. Plasma level of NSE has not been assessed in ALS but is reportedly elevated in other neurological diseases [99,100,101].

The neurotrophin receptor (p75) is a dynamically expressed receptor that is upregulated following neuronal injury [102]. In experimental models of peripheral nerve injury, the extracellular domain of p75 (p75^ECD^) is detectable in urine due to significant upregulation and shedding. It is upregulated in motor neurons from postmortem samples of ALS patients [103]. The receptor’s extracellular domain is cleaved and then passively secreted in urine in ALS and other NDs [102,104]. In ALS, P75^ECD^ has been measured in urine using ELISA [102,105]. Urinary P75^ECD^ levels are significantly more elevated in ALS compared to HCs [105] and NDCs [102]. Additionally, urinary P75^ECD^ levels are positively correlated with the rate of disease progression [102,105] and stage of the disease [105]. Furthermore, they negatively correlate with the ALSFRS-R [105]. P75^ECD^ is readily detectable in plasma [105], but its level has not been studied in ALS.

The most studied candidate biomarkers in ALS are the neurofilament light chain (NfL) and phosphorylated neurofilament heavy chain (pNFH). Neurofilaments (NFs) are cytoskeletal proteins composed of NfL, NF medium chain, and NFH combined with α-internexin in the CNS and peripherin in the peripheral nervous system (PNS) [106.107]. They are largely expressed in the large Betz cells of the motor cortex and myelinated axons as they provide structural support for myelinated axons [106,107]. They are passively released into CSF following axonal injury, irrespective of the primary causal insult [108,109]. Elevated levels of NF are seen in neurological diseases in which there is axonal injury of myelinated neurons [110,111,112,113]. In ALS, NFs have been detectable in serum using ELISA [111,114,115,116,117,118,119,120,121], Simoa [61,117,122,123,124], and ELICA [125,126]. Additionally, they have been detectable in CSF using ELISA [110,113,115,116,117,118,127,128,129], Simoa [94,122,123], ELICA [124,125], the multiplex method [126], and MSM [130]. CSF levels of NfL are significantly more elevated in ALS compared to HCs [112,115,122,124,125,128,129,131,132] and NDCs [106,112,129,132]. Compared to other NDs, plasma NfL levels in ALS are only exceeded by Creutzfeldt–Jakob disease (CJD) [116]. Similarly, levels of plasma NfH are significantly more elevated in ALS compared to HCs [113,129,133] and NDCs [127,129,134].

Additionally, plasma and CSF NfL can differentiate ALS patients from ALS mimics and other motor neuron diseases (MNDs), highlighting its ability to assist clinical diagnosis [114,124,128,129,130,131]. Similarly, CSF NfH was able to differentiate ALS from disease mimics in two large cohorts with high sensitivity and specificity [114,132] but could not differentiate ALS from other MNDs in a recent large cohort despite it being able to differentiate ALS from disease mimics and performing significantly better than plasma NfH [118]. Interestingly, serum NfL and pNfH levels are significantly lower in presymptomatic fALS than in symptomatic ALS and are reportedly elevated 12 to 18 months prior to the emergence of symptoms [120,126] or early symptom onset [133].

NFs concentration is positively correlated with UMN burden [107,111,115,116,121,129,132,135], bulbar onset [114,119,121] rate of disease progression [61,111,115,116,117,119,123,124,129,130,132] and stage of the disease as per the number of involved regions [61,124,129,132]. Additionally, baseline NfL level is negatively correlated with survival [115,116,118,120,122,123,124,127,132,135], the ALSFRS-R [114,116,119,123,130,136] and cognitive performance [136]. In addition to their prognostic potential, NfL [117,124,125,126] and pNFH [130] are relatively stable longitudinally, making them candidate pharmacodynamic biomarkers. Compared to plasma NfH, however, plasma NfL has a steadier trajectory corresponding with disease progression. In addition, inconsistencies in plasma NfH measurements have been reported due to analyte aggregation in what is known as the “hook effect”. This effect was not reported in plasma NfL measurements, which makes plasma NfL a more suitable prognostic and pharmacodynamic biomarker than plasma NfH [125]. In the aforementioned placebo-controlled VALOR trial investigating the use of the intrathecally administered ASO (tofersen) in *SOD1*-caused ALS, there was a significant reduction in CSF NfL levels in the treatment arm compared to the placebo arm. However, as with SOD1, reduction in NfL level was not met with clinical improvement as there was no significant improvement in the clinical endpoints between the two arms of the study [83].

Tau is an axonal microtubule-associated protein involved in structural support and axonal transport and is highly expressed in the axons of CNS neurons [137]. Phosphorylation of tau by specific kinases to phospho-tau (p-tau) leads to its separation from tubulin [138]. Tau is detectable in the CSF after being passively released into the CSF following axonal injury [139,140]. Increased CSF tau levels are reported in several neurological diseases in which there is axonal injury, most importantly AD [139,141]. In tauopathies, tau aggregation has dual anti-inflammatory and proinflammatory effects, but its role in ALS pathogenesis is debatable [142,143].

In ALS, tau proteins have been detectable in CSF using ELISA [8,78,107,144,145,146], Luminex [147], Simoa [94], and CLIA [148]. Studies investigating CSF levels of p-tau and t-tau yielded discrepant results in ALS. While some reported comparable levels between patients and controls [139], others reported increased t-tau levels [147,148,149,150] and a reduction of p/t-Tau ratio [106,144,146]. The mechanism behind a decreased p/t-Tau ratio is unclear [150]. The diagnostic performance of CSF t-tau and p-tau is improved when combined with TDP-43 with an AUC of 0.97 [145]. CSF levels of p/t-Tau ratio are correlated with the ALSFRS-R [8,147], cognitive performance [147], UMN involvement [106,138,147], and rate of progression [94,144].

### 3.3. Biomarkers Related to Neuroinflammation

Deregulated inflammatory response is a key pathological hallmark in ALS, highlighted by the recent discovery of TANK-binding kinase 1’s involvement in the ALS/FTD spectrum [151]. ALS glial cells exhibit proteinaceous inclusions similar to those found in neuronal cells. Those inclusions activate glial cells via chromogranin A and B-mediated pathways [58]. Additionally, the release of cytotoxic and inflammatory mediators from neuronal damage further activates microglial cells. Activated microglia trigger astrocyte activation and disruption of the BBB. BBB disruption allows the infiltration of systemic proinflammatory cytokines into the CNS [152]. Hence, activated microglia promote the secretion of proinflammatory cytokines via a feedforward mechanism [153]. This may initially be a homeostatic response, but altered dynamic motility and phagocytic ability of activated microglia ultimately perpetuate a cycle of neurotoxicity [154]. This is highlighted by experimental evidence in which ablating glial activation in SOD1 disease models extends their survival and improves their function [153,155]. Postmortem ALS tissues illustrate the proliferation of microglia and astroglia in central and peripheral motor neurons [156], with a correlation to reported motor deficits [157]. Despite clear evidence of neuroinflammation as a critical mechanism in ALS neurodegeneration, anti-inflammatory therapies have not been productive [158]. A possible explanation for the failure of anti-inflammatory therapies to suppress neurodegeneration is that microglia are dynamic cells that exhibit different phenotypes, “M1” proinflammatory phenotype or “M2” protective phenotype [155,158].

Astrogliosis is associated with an increased expression of the glial fibrillary acidic protein (GFAP) [159]. GFAP is an intermediate filament-III structural protein found in CNS astroglia, non-myelinating Schwann cells, and enteric glial cells [160]. GFAP assists in the maintenance of astrocyte shape and motility. Additionally, GFAP is involved in the myelination of neuronal axons and the stabilization of the BBB [161,162]. GFAP is passively secreted into the extracellular space upon astrogliosis. Normally, serum levels of GFAP are below the detection limits of available immunoassays. However, GFAP is readily measured in serum in cases of astrogliosis and breakdown of the BBB [161]. In ALS, GFAP has been detectable in CSF and plasma using ELISA [163] and Simoa [164]. CSF and serum GFAP levels are significantly more elevated in symptomatic ALS compared to HCs [156,162] and presymptomatic fALS [163]. ALS patients exhibit over 50% elevation of CSF GFAP levels compared to NDCs [156]. Although GFAP levels do not seem to be significantly correlated with disease progression, they are negatively correlated with cognitive performance and survival [164].

Another protein upregulated in astrogliosis is S100B, which is a subunit of the dimeric protein S100 [165]. S100B is a calcium-binding cytosolic cytokine present in high concentrations in astrocytes and Schwann cells. Extracerebral production of S-100b is minimal, accounting for less than 10% of its plasma concentration, making it primarily brain-specific [166]. S-100b has many intracellular and extracellular functions in the CNS [167]. Under physiological conditions, low nanomolar concentrations of S-100b regulate enzymatic activity and calcium homeostasis and promote the growth of neurites and astrocyte proliferation [168,169]. Pathological conditions of the CNS are associated with increased release of S100B from astrocytes at micromolar concentrations, which have been found to induce apoptosis [165,170]. This may be due to the induction of interleukin-6 secretion and nitric oxide production or through a receptor for advanced glycation end product [RAGE]-mediated apoptosis [171]. S100B is upregulated by astrogliosis or glial damage through mechanisms delineated by previous research [168,172]. Therefore, S-100 B has a dual role as a marker of either glial damage or glial activation [167]. Variable CNS pathologies are associated with elevated CSF levels of S100B [173,174], whereas elevated serum levels of S-100B reflect the breakdown of the BBB [165,175]. In ALS, S100B has been detectable in CSF [176] and serum [177] using ELISA. In ALS, serum S100B is inferior to NfL in supporting ALS diagnosis [177]. Still, S100B may have a prognostic potential as its CSF levels were found to correlate with survival [176] and disease progression [177].

Another RAGE-binding protein implicated in the pathogenesis of ALS is the High Mobility Group Box 1 (HMGB1) protein [178]. HMGB1 is a nuclear protein expressed in several cells, including neurons and microglia [179]. It serves many functions, including regulation of autophagy and stress response [180]. In the CNS, HMGB1 is a byproduct of astrogliosis that stimulates astrogliosis in a feed-forward mechanism [181]. Exposure of astrocytes to interleukins, specifically IL-1β, translocates HMGB1 from the nucleus to the cytoplasm, followed by release as a damage-associated molecular patterns (DAMP) signal [182]. HMGB1 is then passively released from necrotic cells or actively released by proinflammatory cells [180]. The release of HMGB1 activates RAGE, leading to further inflammation and astrogliosis [182]. HMGB1’s role in the pathogenesis of ALS is highlighted by experimental evidence of delayed disease progression after inhibition of HMGB1 receptor signaling [183]. Therefore, HMGB1 not only has potential as a biomarker but also as a therapeutic target in ALS. Postmortem studies show increased expression of HMGB1 in the spinal cord tissues of patients with ALS [184]. Using ELISA, levels of serum HMGB1 autoantibody are significantly more elevated in ALS patients compared to HCs and NDCs [184].

The soluble cluster of differentiation 14 (sCD14) is an emerging biomarker of inflammation. sCD14 is a truncated form of the cell surface glycoprotein expressed by cells involved in innate immunity. The soluble fraction of CD14 is cleaved by plasma protease during inflammatory conditions [185]. sCD14 has a proinflammatory function via activation of the toll-like receptor 4-specific proinflammatory signaling cascade [186]. CSF levels of sCD14 are elevated in ALS patients compared to NDCs and are correlated positively with the rate of disease progression [187] and negatively with survival [176]. The elevation of plasma sCD14 in several inflammatory disorders [188] limits its utility as a biomarker for an ND.

Another implicated factor in ALS-related neuroinflammation is the triggering receptor expressed on myeloid cells 2 (TREM2). TREM2 is a transmembrane glycoprotein that can mobilize across the plasma membrane in microglia in response to ionomycin or interferon-γ [189]. TREM2 is cleaved from the surface of microglia by either phagocytic receptor recycling or ectodomain shedding [190]. TREM2 has been shown to stimulate astrogliosis in various NDs [191,192]. The role of TREM2 in ALS is complex, with both proinflammatory and anti-inflammatory functions [193]. CSF concentration of TREM2 has been measured in ALS using ELISA [194,195]. TREM2 CSF levels are significantly more elevated in ALS compared to HCs [194] and NDCs [195]. TREM2 levels are negatively associated with the rate of disease progression [195] and positively correlated with survival, suggesting a potential neuroprotective effect of TREM2 in ALS [193,195]. However, further supportive evidence is needed.

In the CNS, chitinases are enzymes synthesized by activated macrophages. Chitotriosidase (CHIT1) is produced by activated macrophages and epithelial cells, while chitinase-3-like protein 1 (YKL-40) is produced by activated microglia and reactive astrocytes [196]. They are upregulated in the CNS in neuroinflammatory and NDs, including ALS [197,198,199]. CSF and plasma levels CHIT1 and YKL-40 have been measured in ALS using ELISA [200,201] and LC/MS [202] and appear to be longitudinally stable [201]. CSF levels of CHIT1 and YKL-40 are significantly more elevated in symptomatic ALS compared to HCs [136,201,203] and disease mimics [135,201,203]. Still, the diagnostic performance of CSF levels of CHIT1 and YKL-40 is inferior to that of NfL in two cohorts that included disease mimics [198,203]. CSF levels of CHIT1 and YKL-40 are significantly more elevated in symptomatic ALS compared to presymptomatic fALS [163,201]. Additionally, CHIT1 and YKL-40 are similar between sALS and symptomatic fALS, indicating that their synthesis and release from microglia into the CSF is linked to the symptomatic phase of the disease [163,204]. However, no such association was found in plasma [163]. CSF levels of YKL-40 are positively correlated with disease progression rate [135,136,200,202], NfL levels [200], and pNFH levels [201,202]. CHIT1 and YKL-40 levels are negatively correlated with ALSFRS-R [200] and survival [135,136,198]. CSF YKL-40 levels are negatively correlated with cognitive performance [202].

### 3.4. Biomarkers Related to Blood–Brain Barrier Disruption

It is hypothesized that neuroinflammation causes disruption of the BBB, which in turn allows for the diffusion of inflammatory cells, thereby stimulating inflammation in a feed-forward mechanism [205], further perpetuating the inflammation within the CNS [206]. BBB disruption is evident in all neuroinflammation-mediated NDs, including ALS [206,207]. Postmortem ALS tissues show evidence of BBB disruption in the form of endothelial cell degeneration, capillary leakage, and perivascular edema [208]. ALS patients have significantly increased CNS perivascular hemoglobin deposits, as evidenced by immunostaining for glycophorin a, indicating erythrocyte extravasation. In ALS models, BBB disruption and erythrocyte extravasation precede the onset of weakness [209]. The permeability of the BBB can be indirectly assessed via CSF protein content and the CSF/plasma albumin quotient (Q-Alb) [210]. Studies on BBB integrity in ALS patients have been conflicting. While some found significant elevations in CSF Q-Alb levels in ALS patients compared to NDCs [211,212], others did not [213]. The discrepancy in the findings of the aforementioned studies might be attributed to the types of NDCs included, as the latter study included patients with conditions associated with BBB disruption [213] while the other studies did not [211,212]. Similarly, the prognostic value of CSF Q-Alb is debatable. While some found the Q-Alb to be negatively correlated with survival in males [213,214] and both sexes [211,212], others did not find a significant correlation [215,216]. Interestingly, the Q-Alb has been found to be elevated in diabetes mellitus, which might be attributable to associated microvascular changes [213]. Therefore, it is important to document comorbidities which may affect the levels of the biomarkers. Indeed, patients with ALS-FTD have higher levels of Q-Alb than ALS patients with no evidence of FTD, suggesting that BBB disruption may be more significant in the former group [212].

### 3.5. Biomarkers Related to Syanptopathy

Alterations of synaptic structure and function, known as synaptopathy, are some of the earliest abnormalities seen in ALS disease models, often preceding neuronal loss [217]. Postmortem samples from ALS patients show dendritic spine loss in the corticospinal motor neurons [218]. Though synaptopathy is primarily linked to excitotoxicity, it is likely to be the result of several converging mechanisms, such as the accumulation of misfolded proteins, mitochondrial dysfunction at distal axon terminals, and microglial activation [217,219]. Conversely, synaptopathy propagates mitochondrial dysfunction, aberrant proteostasis, and defective neuromuscular junctions [220]. Excitotoxicity is attributed to increased glutamate release from presynaptic neurons, alterations in post-synaptic glutamate receptors, and dysfunction of EAAT2 [221,222]. ALS disease models exhibit reduced EAAT2 expression and subsequent induction of reactive oxygen species (ROSs), which further disrupts glutamate uptake by astrocytes [223]. Postmortem studies show altered density of post-synaptic glutamate receptors and glutamate re-uptake binding sites in the spinal cord of ALS patients, resulting in elevation of synaptic glutamate [224]. It has been recently suggested that the expression of the AMPAR subunit GluR2 is regulated by a microRNA (miRNA) pathway. However, the exact molecular mechanism underlying the regulation of receptor signaling and expression has yet to be explored [225]. Elevated levels of the excitatory neurotransmitter glutamate increase calcium entry into neurons, thereby inducing neurodegeneration through calcium-dependent enzymatic pathways [139,226]. Motor neurons are particularly susceptible to excitotoxic injury due to low levels of calcium-binding proteins and subsequent low calcium-buffering capacity [227]. Calcium overload disrupts mitochondrial homeostasis, leading to energy depletion, as well as activation of enzymes, such as calpains, resulting in the degradation of key cellular molecules [219]. Excitotoxicity is targeted by the one globally approved drug for ALS, riluzole. By inhibiting glutamate release, riluzole blocks persistent sodium currents and reduces neuronal excitability, achieving modest improvement in life expectancy [221].

CSF glutamate has been studied in ALS using high-performance liquid chromatography [226,228] and ion-exchange chromatography [224]. In a study of 29 ALS patients, 28 had elevated CSF glutamate, but this had no association with disease progression [226]. In a larger cohort of 377 patients, 40.8% of patients had elevated CSF glutamate levels with a correlation to spinal onset and disease progression. The researchers hypothesized that the correlation with the site of onset is likely related to the size of the injury [228]. Although both studies used high-performance liquid chromatography, the differences in the percentages might be due to the different sample sizes or the fact that all patients in the first study had spinal onset [226], while 71.4% of patients in the latter study had spinal onset [228]. In a study using ion exchange chromatography in 37 ALS patients, 61% had CSF glutamate levels within the normal range despite there being significant elevation in the ALS group compared to NDCs. It is worth noting that in this study, 12 patients had bulbar onset, but this had no correlation to CSF glutamate levels. The study showed heterogeneous levels of CSF glutamate in ALS patients with no significant clinical correlation. The authors warranted further investigation into the effect of heterogeneity of CSF glutamate levels on response to anti-glutamate therapy. The same study performed a postmortem analysis of spinal cord tissue obtained from seven deceased participant ants, illustrating a significant negative correlation between CSF glutamate level and the density of 3H]D-aspartate binding sites. The authors suggested that a defect of presynaptic glutamate transport may underlie elevated CSF glutamate [224]. This is supported by another postmortem study illustrating reduced concentration of the glial glutamate transporter protein (GLT-1) in ALS motor cortex tissue [229].

When interpreting biofluid glutamate data, it is important to consider methodological and biochemical differences. The accuracy of CSF glutamate levels is likely affected by acidification with perchloric acid and its subsequent neutralization with potassium carbonate [230], which was performed to improve stability by [228] but not [224,226]. Additionally, CSF levels of glutamate may be inaccurately elevated in cases where there is stagnation of CSF in the lumbar theca with formation of glutamate from degradation of glutamine or proteins [230] as in cases of lumbar canal stenosis [224].

The results from studies on plasma glutamate levels are conflicting. Using ^1^H NMR spectroscopy [231] and gas chromatography-MS [232], serum glutamate was found to be significantly more elevated in ALS patients compared to HCs. Additionally, serum glutamate was positively correlated with disease duration [231] and disease progression [232]. It was not determined whether patients’ plasma samples were collected after fasting [231,232] as plasma glutamate levels may be affected by fasting [233]. Using ion-exchange chromatography, there was no statistically significant difference in serum glutamate levels between ALS patients and NDCs [224]. In fact, another study using ^1^H NMR spectroscopy reported that NDCs had significantly higher plasma glutamate levels than ALS patients [231].

Other biomarkers of synaptopathy have not been as well studied in ALS as they have been in other NDs, such as neurogranin (Ng) and visinin-like protein 1 (VILIP 1). Elevated circulating levels of both proteins and decreased concentrations in brain tissue in those areas may reflect the intensity of synaptic degeneration [234]. Ng is a small post-synaptic substrate for protein kinase C and is abundantly expressed in the dendritic spines of the telencephalon [235]. Some Ng peptides are specific for CSF, while others are specific for plasma, making uniform detection techniques more challenging [236]. Using bead-based immunoassay, AD patients had markedly elevated CSF Ng levels compared to other NDs, including ALS [237]. VILIP-1 is a neuronal calcium-sensor protein expressed in neuronal dendrites of the telencephalon and is involved in synaptic plasticity, calcium homeostasis, as well as other functions [238]. Using Simoa, a study evaluated CSF and serum VILIP-1 as a biomarker for AD compared to other NDs. Patients with CJD had remarkably higher levels than any other ND, including AD, while VILIP-1 levels in ALS patients were comparable to those in HCs [238]. A possible explanation for lower Ng and VILIP-1 levels in ALS compared to AD is that both proteins are abundantly expressed in brain areas crucial for cognitive function [239,240].

### 3.6. Biomarkers Related to Oxidative Stress

Oxidative stress is a key mechanism in ALS neurodegeneration, as highlighted by experimental studies and postmortem studies [241,242]. fALS caused by SOD1 mutations further implicates oxidative stress in ALS pathogenesis [243]. Oxidative stress causes the mitochondrial aconitase to aggregate in the mitochondrial matrix, causing mitochondrial dysfunction, which in turn amplifies oxidative stress [244,245]. Additionally, released ROSs impair glial uptake of glutamate, resulting in glutamate-induced calcium overload [227]. Increased mitochondrial susceptibility to calcium overload disrupts mitochondrial ATP synthesis and its buffering capacity, further escalating calcium overload [246]. Together, mitochondrial dysfunction and calcium overload interrupt the binding of kinesin-1 to tubulin, which compromises the microtubule stability and, subsequently, axonal transport [27]. In addition, oxidative stress induces glial activation and the production of proinflammatory cytokines and other ROS [227]. Given their high metabolic rate and their limited ability for regeneration, motor neurons are particularly susceptible to stress-induced neurodegeneration [20]. Oxidative stress is the therapeutic target of the FDA-approved edaravone [247]. Edaravone has been shown to reduce disease progression, most notably in patients whose forced vital capacity (FVC) is ≥80% [248]. However, the failure of antioxidants to halt disease progression highlights the interlinked nature of the several pathophysiological mechanisms implicated in ALS [26,242].

Postmortem ALS studies show evidence of increased expression of 8-hydroxy-2′-deoxyguanosine (8OH2′dG), which is a DNA base adduct generated from guanine by hydroxyl radical damage and serves as a measure of oxidative damage to DNA [249,250]. Similarly, there is an increased expression of 8-isoprostane (IsoP), a peroxidation product of arachidonic acid [251]. Increased production of both oxidation products results in increased secretion in the urine [250,251]. 8OH2′dG and IsoP have been measured in ALS using ELISA in CSF and plasma [249] as well as in urine using ELISA [249,252] and liquid chromatography [252]. 8OH2′dG is significantly more elevated in ALS patients compared to HCs, whether measured in CSF, plasma, or urine, with a positive correlation with disease progression [249]. Urine 8OH2′dG and IsoP are significantly more elevated in ALS patients compared to HCs, with a positive correlation with age in both groups, indicating that oxidative stress is age-associated [252].

Another product of lipid peroxidation is 4-Hydroxynonenal (HNE), which functions as a second messenger of oxidative/electrophilic stress [253]. Under physiological conditions, NHE is involved in antioxidant defense by stimulating the removal of damaged cellular components. Under pathological levels of oxidative stress, there is excessive upregulation of HNE [253]. Experimental evidence illustrates HNE’s role in oxidative stress signaling and stimulation of intrinsic and extrinsic apoptosis, highlighting its potential role as a therapeutic target [254]. Application of HNE in motor neuron cultures results in levels of oxidative stress consistent with those seen in ALS [255]. This highlights the active role of HNE in oxidative stress in addition to its biomarker potential. CSF and plasma HNE have been detected in ALS using liquid chromatography and ELISA [255,256]. CSF HNE levels are significantly more elevated in ALS compared to HCs [256] and NDCs [255] and appear to correlate with the stage of the disease as per the Appel ALS score, though not with the rate of disease progression [256].

Uric acid (UA) is the final oxidation product of purine metabolism and a free-radical scavenger essential for redox homeostasis [257,258]. Theoretically, high concentrations of UA can prevent stress-induced neurodegeneration [259]. Application of UA on ischemic neuronal cell cultures prevents oxidative stress [260]. Additionally, elevated levels of serum UA are associated with a lower risk and slower progression in various NDs [261,262]. ALS patients have significantly decreased levels of plasma UA compared to HCs [259,263,264], even in early disease [265]. However, the use of UA to assist diagnosis is limited by the alternation of its levels in various neurological and non-neurological conditions [261,262,266]. In ALS, UA levels are negatively correlated with disease progression rate [257,259,265] and positively correlated with FVC [264], cognitive performance [267] and survival [265]. There is a correlation between lower UA levels and bulbar onset. However, whether this is related to the malnutrition imposed by dysphagia is not clear [268]. In one study, the seemingly protective effect of UA in ALS, adjusted for onset, was only significant in male patients [262]. High levels of CSF UA have been reported in some neurological disorders [269], but to the best of the authors’ knowledge, there are no published data regarding CSF levels of UA in ALS.

### 3.7. Biomarkers Related to Aberrant RNA Processing

A remarkable number of proteins implicated in ALS pathogenesis are linked to RNA processing. The predominant protein involved in ALS pathogenesis, TDP-43, is involved in the processing of RNAs and miRNAs [24]. Both are important epigenetic regulators of transcriptome plasticity essential for the survival of mature motor neurons [270]. Mutations in *SOD1, C9ORF72*, and *FUS* genes alter RNA metabolism directly or indirectly [67,271]. Additionally, toxic aggregates alter RNA splicing, capping, polyadenylation, and transport [27]. Toxic aggregates also disrupt miRNA biogenesis via altering stress granule dynamics. This reinforces the role of aberrant RNA processing and miRNA biology as critical mechanisms in ALS pathogenesis [272]. MiRNAs are small non-coding RNAs that post-transcriptionally regulate protein-coding mRNAs via binding to complementary regions of targeted mRNA and governing the translation and degradation of target mRNAs involved in different biological functions. The expression of various miRNAs in the CNS is believed to play a vital role in neuronal development [273]. Accumulating evidence recognizes deregulation of miRNA processing as a key mechanism in ALS pathogenesis and a potential therapeutic target [273]. Experimental models that do not process miRNA in motor neurons exhibit signs of muscle denervation [274]. Additionally, postmortem ALS motor neurons exhibit deregulation of various miRNAs [270]. Deregulation of miRNA is due to biogenesis defects or miRNA transcriptional changes [275]. Expression of circulating miRNA in ALS is heterogeneous, with reported upregulation of some miRNAs and downregulation of others. *SOD1* disease models show upregulation of miR-125b which has a proinflammatory effect through promoting NF-κB signaling in microglia [39]. Conversely, *SOD1* disease models show downregulation of miR-106b-25, which normally regulates mediators of apoptosis [276,277]. Upregulation of pre-miR-129-1 has been shown to inhibit neurite outgrowth and differentiation, whereas inhibition of pre-miR-129-1 improves the survival of *SOD1* models [273]. Based on experimental and clinical evidence from mutation carriers, this is believed to occur in the presymptomatic phase of ALS [272,278].

MiRNAs are the most extensively studied class of small-noncoding RNAs. They are packed into extracellular vesicles, attached to proteins or free in biofluids from which they can be measured with high stability due to their immunity to RNase degradation and ability to withstand different temperatures and pH conditions [270,272,279,280,281]. Whether circulating miRNA molecules are passive degradation by-products of an underlying pathology [282] or are genuinely active molecules [281] is debatable. Measurements in humans are usually carried out after profiling biofluid samples from ALS disease models [277] or profiling miRNAs known to regulate ALS-related genes [278]. Levels of different miRNAs have been measured in ALS using real-time quantitative polymerase chain reaction (RT-qPCR) [270,283,284,285] and are longitudinally stable [286]. In the CSF, ALS patients show significantly lower expression levels of miR-132-5p, miR-132-3p, miR-143-3p and increased levels of miR-143-5p, miR-574-5p compared to HCs [271]. In the serum, fALS patients show significantly lower expression levels of miR1915-3p, miR3665, miR4530, and miR4745- 5p [287], while sALS patients show decreased expression of miR1234- 3p and miR1825 [284] in two studies profiling the same set of miRNAs, suggesting potential differential epigenetic deregulation between fALS and sALS [270]. In a large cohort of 252 ALS patients, plasma samples from patients have significantly increased levels of the predominantly neuronal miRNA (miR-181) compared to HCs. Additionally, plasma levels of miR-181 are negatively correlated with survival and have a comparable prognostic value to that of NfL [286]. As miR-181 is harbored in neuronal axons, it is thought that its elevated levels are due to passive release from axonal degeneration similarly to NFs [286]. Another predominantly neuronal miRNA is miR-338-3p, which targets the principal transporter of glutamate (SLC1A2) and acts as indirect modulator of apoptosis through regulation of AATK mRNA levels. MiR-338-3p is significantly overexpressed in serum and CSF of ALS patients compared to HCs and ND controls and has been shown to correlate with disease duration [278]. Another study profiled 37 neuronal miRNAs found the following ratios (miR206/miR338-3p, miR9/miR129-3p, and miR335-5p/miR338-3p) to differentiate ALS from other NDs with a sensitivity of 84% and a specificity of 82%. Interestingly, the study reported differential miRNA profiles in males and females outside of the aforementioned ratios. Reported gender differences in neuronal miRNA regulation warrant further investigation, especially since males, who are incidentally less likely to have a bulbar onset [288], represented 72% of the sample [289]. Onset-related differences were reported in another study that found that seven of 38 downregulated systemic miRNAs (miR30B-5p, miR30C-5p, miR106B-3p, miR128-3p, miR148B-3p, miR186-5p, miR342- 3p) in blood had significantly lower expression in spinal onset compared to bulbar onset [290]. Similarly, lower expression of the proinflammatory glial miR-155 was found to be significant in spinal onset ALS [291]. The molecular mechanisms underlying and resulting from these phenotype alternations are unclear.

Deregulated miRNA profile in ALS disease models, including *SOD1* transgenic mice and *TDP-43* mice, is not always reflected in patients. In one study, only two of six deregulated serum miRNAs in mice were deregulated in ALS patients. The two miRNAs (miR142-3p and miR1249-3p) were significantly altered in ALS patients compared to HCs. Additionally, miR-142-3p was negatively correlated with the ALSFRS-R and has been hypothesized to alter the expression of TDP-43 and C9orf72 based on a bioinformatics analysis from the same study [272]. Still, deregulation of miR-142-3p expression is observed in other NDs [292]. Although the physiological function of miR-142-3p is unknown, experimental models of neuroinflammation show that it mediates IL-1β-dependent downregulation of the glial glutamate-aspartate transporter (GLAST), thus contributing to excitotoxicity [285]. It may be deduced that upregulation of miR-142-3p in ALS patients is associated with neuroinflammation. However, this requires further research. Interestingly, miR-1249-3p is upregulated in serum samples from *SOD1* models but significantly downregulated in ALS patients compared to HCs [272]. Although little is known about the function of miR-1249-3p, it has been shown to be deregulated in several disorders [293]. Despite miR-1249-3p showing significant differences between ALS patients and HCs, the discordant findings between the disease models and ALS patients highlight the restriction of ALS disease models to reflect the transcriptome of human ALS [294].

Extensive deregulation of miRNA profile in ALS patients has been reported by several other studies affecting miR-206, miR-143-3p, miR-374b-5p [282}, miR-129-5p [273], miR-1234-3p, miR-1825 [284], miR181a-5p, miR21-5p, miR15b-5p [33], miR-146a and miR-149 [291], summarized in Figure 2. However, a highly heterogeneous miRNA profile with little overlap is seen between the findings of the studies, which may be attributable to technical differences such as alternative RNA isolation techniques [270,280,281] or, indeed, the nature of miRNA expression. Although miRNA molecules are remarkably stable in biofluids [33], their expression may be altered by factors unrelated to pathology, such as nutrition and medication [295]. It is therefore important to document the fasting status of participants, which may account for changes in miRNA expression [280]. A negligible correlation is seen between miRNA levels in the CSF and plasma, suggesting different regulatory mechanisms in the two fluid compartments [270,291,296].

### 3.8. Biomarkers Related to Muscle Changes

Nerve terminal susceptibility to oxidative stress and calcium overload accelerates presynaptic failure and disrupts neurotransmitter release from the presynaptic terminals. Defective acetylcholine (Ach) handling in presynaptic terminals likely contributes to the degeneration of distal axons [32]. Muscle involvement in ALS is largely thought to be secondary to degeneration of alpha motor neurons, distal axons, and neuromuscular junctions [297]. With ongoing axonal degeneration, denervated muscle fibers may be reinnervated through collateral sprouting from unaffected populations [298,299]. Denervation is observed prior to symptom onset and becomes symptomatic after an estimated loss of over 50% of motor units [298]. There is seemingly an order by which motor units are lost, which is highlighted in a previous article [299]. However, researchers now contest the previously assigned bystander status of muscle in ALS, suggesting that muscle defects occur independently of motor neuron degeneration, and while denervation inarguably affects muscle [300], muscle fibers in ALS are more severely affected by denervation than normal muscle fibers as muscle fibers in ALS have reduced Ach affinity [32]. Muscle mitochondria uncoupling [301] and aberrant expression of myogenic regulatory factors such as muscle-specific miRNAs [291,302] have been suggested as independent intrinsic muscle pathogenic mechanisms.

Muscle-specific miRNAs (myomas) MiR-1, miR-206, miR-133a, miR-133b, and miR-27a are involved in skeletal muscle proliferation, differentiation, and regeneration and are considered markers of residual muscle mass [291]. MiR-206 and miR-133b are proposed to have a protective role in ALS as they promote the regeneration of functional synapses in response to injury [291,303]. Conversely, some miRNAs, such as miR-143-3p and miR-374b have been shown to suppress myoblast cell differentiation [304]. In a study of fourteen ALS patients, the expression of miR-206 and miR-133 in serum was significantly increased, while that of miR-27a was significantly decreased compared to HCs. Additionally, in the four patients with bulbar onset ALS, there was a significant down-regulation in serum miR-133a, miR-133b, and miR-206 compared to spinal onset. The authors attributed differential expression of myomiRs to faster disease progression in bulbar ALS. The study also reported significant muscle atrophy in muscle fiber morphometric analysis in bulbar ALS compared to spinal ALS [291]. However, a longitudinal study found that levels of miR-143-3p and miR-374b-5p but not MiR-206 levels correlated with disease progression, questioning its value as a prognostic biomarker. Additionally, there was no difference in the expression of the three aforementioned miRNAs between the riluzole-treated and untreated groups. There was a longitudinal increase in the expression of miR-143-3p and a decrease in the expression of miR-374b [280]. In cell culture, over-expression of both miRNAs suppresses differentiation of C2C12, making their opposing expression unclear [304]. The authors hypothesized that reduced expression of miR-374b-5p may be a compensatory response to the degeneration of muscle in an effort to restore myoblast differentiation and muscle regeneration. It is unclear, however, why the expression of miR-143-3p does not similarly drop, but the authors suggest that it may be a mere by-product of muscle denervation [280]. However, this is debatable given the negative correlation between miR-143-3p expression level and level of serum creatinine kinase [272]. Despite its lack of prognostic potential and lack of specificity, MiR-206 is one of a few miRNAs that has been identified in multiple studies [280,282,291,305]. MiR-206, a regulator of histone deacetylase 4, is a candidate biomarker that may be used to assist in the diagnosis of ALS. It is abundant in skeletal muscle and has been hypothesized to be released into the circulation as a passive by-product of muscle injury [305].

Interestingly, the levels of myomiRs have been found to drop after six weeks of moderate aerobic exercise in association with a significant improvement in ALSFRS-R in a preliminary study of eighteen patients. The authors suggested that the changes in circulating myomiRs levels may be attributed to the muscles being in recovery [306]. However, it is worth noting that the study had no control group and that enrollment in an exercise program might be associated with concurrent dietary and therapeutic factors that may alter the expression of miRNAs [295]. Low expression of other miRNAs involved in muscle regulation has been reported by other studies, such as miR-27a [291]. However, no study seemed to assess myomiRs levels in presymptomatic ALS nor in ALS mimics.

Creatinine kinase (CK) is an intracellular enzyme abundantly present in striated muscle and is elevated in case of injury to the sarcolemma, increased muscle cell membrane permeability, or up-regulation in response to increased energy demand [307]. Neuropathological studies in ALS show limited necrosis of muscle fibers, suggesting that CK elevation is not related to the lysis of muscle fibers [308]. CK elevation in ALS might be a manifestation of muscle denervation, which is associated with increased muscle cell membrane permeability [309]. CK levels are also elevated in myopathies, myositis, and rhabdomyolysis [310,311]. Serum CK levels are significantly more elevated in ALS patients compared to HCs [307]. Male gender and spinal onset are independently associated with higher UA levels [307,308]. Serum CK levels are positively correlated with lean body mass and survival [308] and negatively correlated with disease progression [312]. In a large cohort of 237 ALS patients, CK levels correlated with the electromyographic mean spontaneous potential score, supporting the hypothesis that CK elevation is induced by LMN loss and muscle denervation [309]. However, it is unclear whether heterogeneity in CK levels is due to compensatory upregulation of metabolic pathways or inherently different processes in patients with different progression rates [312].

Titin is a sarcomeric protein expressed in striated muscle connecting the Z and M lines in the sarcomere [313]. It is involved in muscle tension and viscoelasticity [314,315]. Sarcomeric disruption in response to muscular damage results in the degradation of titin and the release of its fragments into plasma [315]. The N-terminal titin fragment (N-titin) is released in urine and has been detected in several muscular dystrophies [315,316,317]. Using ELISA, ALS patients have significantly higher urinary N-titin than HCs. On multivariate analysis, urinary N-titin and serum NfL are independent predictors of survival, while urinary p75 ^ECD^ is not [318].

Myostatin, also known as growth differentiation factor 8 (GDF8), is a protein that inhibits the proliferation of satellite cells, muscle differentiation, and muscle growth. Mutations in the myostatin gene enhance myoblast regeneration and promote skeletal muscle hypertrophy [319]. Myostatin is regulated by binding proteins, most notably by follistatin and miRNA-27a, which are negative regulators of myostatin [291,320]. Inhibition of myostatin poses a potential therapeutic target in ALS. Experimental inhibition of myostatin in preclinical models improves muscle mass and function, but its effect on survival is conflicting [321,322]. A clinical trial testing the safety and efficacy of an anti-myostatin antibody is underway in patients with spinal muscular atrophy [323], but no trials have been initiated in ALS. Levels of serum myostatin and its inhibitor follistatin have been tested in ALS patients. Using ELISA, it was found that the ratio of myostatin/follistatin was significantly more elevated in ALS patients than in HCs and was higher in bulbar ALS than in spinal ALS. Additionally, there was a negative correlation between myostatin and miR-27a, which is believed to suppress the expression of myostatin. However, it was not determined whether the ratio of myostatin/follistatin or the expression of miR-27a correlated with clinical status [291].

## 4. Conclusions

ALS is an intricate non-cell autonomous disease whose pathology encompasses multiple levels of the neuromuscular system with several interrelated mechanisms involving different cell types (including interneurons, microglia, and myocytes) through various molecular and genetic pathways. The development of biomarkers for ALS is complicated by its heterogeneity and the involvement of several converging pathogenic mechanisms. A significant number of biomarkers related to ALS pathophysiology can be measured in biofluids using readily available assays, as shown in Table 1. Still, head-to-head comparison between biomarkers is not possible in most studies due to different study designs, different sample sizes, different control groups, and different protocols. This highlights the need for standardization of collection methods, procedures, controls, endpoints, and reporting guidelines tailored to ALS, as there have been for other neurological disorders [66]. Blood biomarkers may be the most pragmatic due to noninvasive sampling and ease of longitudinal measurements, as illustrated in Figure 3. Still, blood is a complex compartment that requires a preemptive understanding of the possible interactions for a potential biomarker and the factors that might alter the levels of a biomarker, such as diet and comorbidities.

Perhaps the most consistent findings in ALS biomarkers involve biomarkers of neurodegeneration, specifically NfL, which may be used to complement clinical diagnosis given its high sensitivity and the availability of its detection techniques. Additionally, NfL has had a consistent prognostic potential and potential utility as a pharmacodynamic biomarker, given its stability over time. Furthermore, as the elevation of NfL predates the emergence of symptoms, it allows the monitoring of the disease in its early stages, which can facilitate early diagnosis and pre-clinical treatment. Despite the presymptomatic elevation of NFs, it is argued that the release of NFs is the end product of the neurodegenerative cascade [324] and may, therefore, reflect a point of irreversible axonal injury. However, apart from mutation biomarkers, no other biomarker has been documented to have been altered prior to the emergence of symptoms. Additionally, NfL can be measured in serum using a variety of techniques, including ELISA and Simoa. Before NfL can be introduced to the clinic, future large-scale prospective studies are needed to establish cutoff points for ALS phenotypes as well as controls specific to the measurement technique. It is hypothesized that the release of NfL is correlated with the UMN burden. However, to establish a clear NfL connection to either UMN degeneration or LMN degeneration, the development of specific assays targeting α-internexin or peripherin may be required to reflect NfL levels related to either degeneration of UMNs or LMNs, respectively [325]. In addition to NfL, a panel of biomarkers may be used to better reflect the heterogeneous pathophysiology of ALS. TDP-43’s key role in ALS pathogenesis is well supported by literature, yet its status as an ALS biomarker is debatable. Reports of limitations of the TDP-43 immunoassays undervalue the TDP-43 centric approach in search of a specific biomarker. Still, plasma TDP-43 had prognostic and diagnostic potential against controls, though not including disease mimics. Despite the seemingly central role of aberrant RNA processing in ALS pathogenesis, there is sparse homogeneity in the reported differentially expressed miRNAs apart from miR-206, which has been identified by several studies despite its lack of specificity or correlation with disease progression. Research on aberrant RNA processing will likely expand in light of the discovery of circulating non-miRNA ncRNA [281].

## Figures and Tables

**Figure 1 ijms-25-10900-f001:**
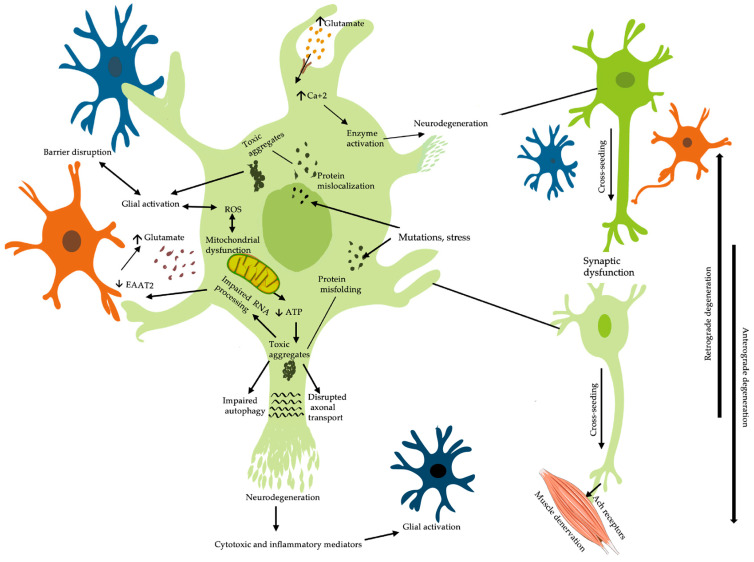
Pathophysiology of amyotrophic lateral sclerosis. The interaction of genetic, epigenetic, and environmental factors triggers mutations and stress conditions, which induce mislocalization or misfolding of intracellular proteins. Mislocalized and misfolded proteins are capable of self-seeding and cross-seeding other proteins into toxic aggregates. These aggregates disrupt axonal transport, mitochondrial respiration, and the clearance of damaged proteins, induce glial activation, and sequester proteins essential for RNA processing. Deregulated processing of the EAAT2 mRNA transcript reduces the expression of the transporter, leading to excessive glutamate concentration, subsequent excessive calcium influx, and overproduction of ROS, which further induce glial activation and creates a continuous cycle of mislocalization and misfolding of the proteins as well as activation of enzymatic pathways which propagate neuronal injury. Calcium overload and oxidative stress disrupt glutamate uptake by astrocytes and axonal transport and contribute to mitochondrial failure, leading to energy depletion, which induces altered proteostasis and the continuation of the cycle. Astrogliosis triggers blood-brain barrier (BBB) disruption and macrophage infiltration, leading to neuroinflammation. These converging mechanisms result in neurodegeneration, which stimulates astrogliosis through released inflammatory and cytotoxic mediators. Neurodegeneration propagates across motor neurons, leading to the disassembly of the neuromuscular junctions and the denervation of skeletal muscles. EAAT2: excitatory amino acid transporter 2; ROS: reactive oxygen species.

**Figure 2 ijms-25-10900-f002:**
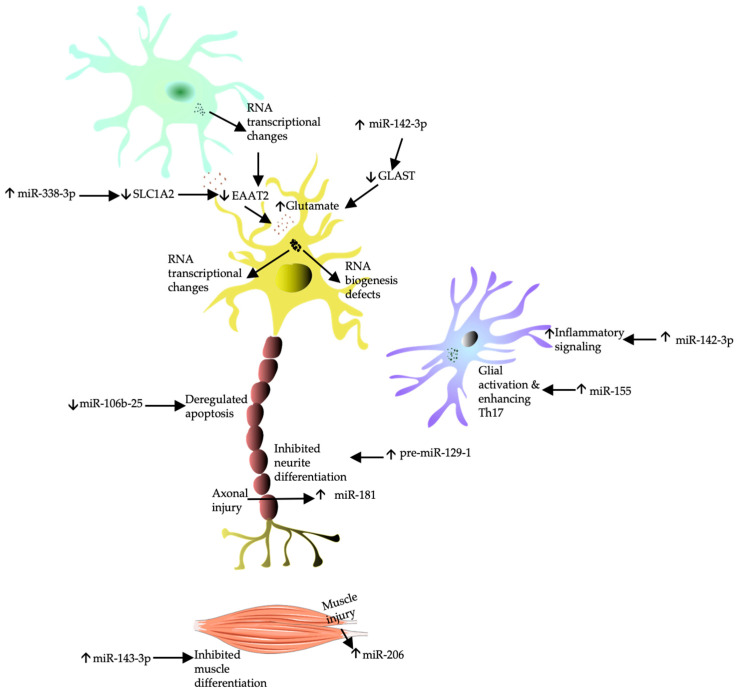
Summary of deregulated RNA profile in amyotrophic lateral sclerosis. Aberrant RNA processing in ALS is attributed to biogenesis defects and/or miRNA transcriptional changes predominantly caused by altered proteostasis as the toxic aggregates alter RNA Splicing, capping, polyadenylation, and transport and disrupt miRNA biogenesis via altering stress granule dynamics. A profile of deregulated miRNAs is illustrated here based on experimental and clinical studies of abnormal levels of several biomarkers linked to neuronal, excitotoxic, inflammatory, and muscle-related mechanisms. EAAT2: excitatory amino acid transporter 2; GLAST: glutamate aspartate transporter; SLC1A2: solute carrier family 1 member 2; Th17: T helper 17 cell.

**Figure 3 ijms-25-10900-f003:**
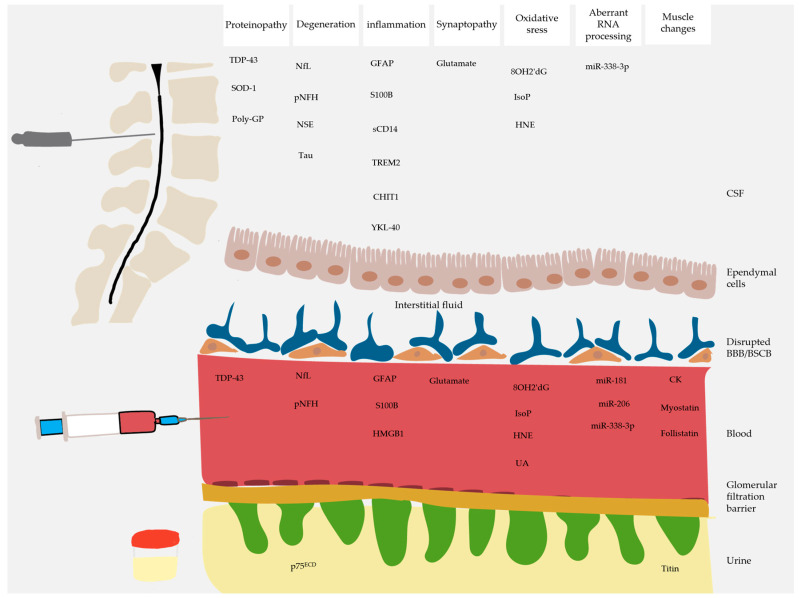
Availability of candidate biomarkers for amyotrophic lateral sclerosis in biofluids. Several pathogenic mechanisms lead to the release of ALS-related biomarkers either passively or actively. Many of these are in high abundance in the CSF from ALS patients, but the invasiveness of CSF sampling limits research recruitment and subsequent longitudinal measurement. Similarly, multiple ALS-related biomarkers can be measured in the blood with less invasive sampling. The availability of biomarkers in the urine is limited by filtration restriction of proteins with high molecular weight. BBB: blood–brain barrier; BSCB: blood–spinal cord barrier; CSF: cerebrospinal fluid.

**Table 1 ijms-25-10900-t001:** Summary of potential fluid biomarkers in amyotrophic lateral sclerosis.

Pathophysiology	Biomarker	Detection Technique	Clinical Findings
Proteinopathy		ELISA [57,58,59,63]	Assists diagnosis vs. controls [57,58,59,60] Prognostic [57,59,61]
	TDP-43	Simoa [60,61]	
		Mass spectrometry [62]	
	SOD1	ELISA [78,81] LC-MS [82]	Assists diagnosis vs. controls [82] Prognostic [83]
	DPRs	MSM [18,91,92] Simoa [93]	Diagnostic of mutation [18,91,92,93]
Neuroinflammation	GFAP	ELISA [163] Simoa [164]	Assists diagnosis vs. controls [156,162] Prognostic [164]
	S100B	ELISA [176,177]	Prognostic [176,177].
	HMGB1 ab	ELISA [184]	Assists diagnosis vs. controls [184]
	sCD14	ELISA [187]	Assists diagnosis vs. controls [187] Prognostic [176,187]
	TREM2	ELISA [194,195]	Assists diagnosis vs. controls [194,195] Prognostic [193,195]
	CHIT1 & YKL-40	ELISA [200,201] LC/MS [202]	Assists diagnosis vs. controls [136,201,203] Assists diagnosis vs. mimics [135,201,203] Prognostic [135,136,198,200,202]
BBB disruption	Q-Alb	Immunoturbidimetric assay [219]	Prognostic [218]
Oxidative stress	8OH2′dG IsoP	ELISA [249,252] Liquid chromatography [252]	Assists diagnosis vs. controls [249,252] Prognostic [249,252]
	HNE	ELISA [255,256] Liquid chromatography [255,256]	Assists diagnosis vs. controls [255,256]
	UA	Liquid chromatography [269]	Assists diagnosis vs. controls [259,263,264,265] Prognostic [257,259,264,265]
Synaptopathy	Glutamate	Liquid chromatography [226,228] Ion-exchange chromatography [224] Gas chromatography-MS [234] (1) H NMR spectroscopy [233]	Assists diagnosis vs controls [224,231] Prognostic [231,232]
Aberrant RNA processing	miR-181	PCR [286]	Assists diagnosis vs. controls and prognostic [286]
	miR-206	PCR [280,282,291,305]	Assists diagnosis vs. controls [280,283,291,305]
	miR-338-3p	PCR [278]	Assists diagnosis vs. controls and prognostic [278].
[278].Neurodegeneration	NSE	ECLIA [98]	Assists diagnosis vs. controls and mimics [98].
	p75^ECD^	ELISA [102,105]	Assists diagnosis vs. controls [102,105] Prognostic [102,105]
	NFs	ELISA [111,114,115,116,117,118,119,120,121,134,135,136] Simoa [61,117,123,124] ELICA [125,126] Multiplex method [126] MSM [130]	Assists diagnosis vs. controls [112,113,115,122,124,125,128,129,131,132,133]. Assists diagnosis vs. mimics [114,118,124,128,129,130,131,132]. Prognostic [61,111,114,115,116,117,119,121,123,124,125,128,129,132,136]
	Tau	ELISA [8,78,107,144,145,146] Luminex [147], Simoa [94] CLIA [148]	Prognostic [8,94,144,147]
Muscle changes	CK	Enzymatic method [307]	Prognostic [312]
	N-titin	ELISA [318]	Assists diagnosis vs. controls and prognostic [318]
	myostatin/follistatin	ELISA [291]	Assists diagnosis vs. controls [291]

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
