# Peer review of "A Review of Biomarkers of Amyotrophic Lateral Sclerosis: A Pathophysiologic Approach"

_ijms, 2024, doi:10.3390/ijms252010900_

Round 1

Reviewer 1 Report (New Reviewer)

Comments and Suggestions for Authors

The review article by Rawiah Alshehri and colleagues provides a detailed review of the pathophysiologic approach of Amyotrophic Lateral Sclerosis (ALS) biomarkers. In this review, the authors have highlighted the heterogeneous nature of ALS at the clinical, genetic, and neuropathological levels. The review has explored ALS pathophysiology, the challenges of developing diagnostic and prognostic tools that fit all disease phenotypes, and the limitations associated with current tools. Overall, this well-written review discusses quantitative biofluid biomarkers and their advantages and disadvantages. The overall quality of the manuscript could be improved, and the following changes are needed in the article.

In section 2, the authors discussed the pathophysiology of ALS (familial cases and sporadic cases) and identified several pathogenic mutations that have revealed various mechanisms and genes involved, such as C9orf72, SOD1, TARDBP, FUS, etc. However, the authors should provide a figure on the pathophysiology of ALS. This should cover the molecular pathways and correlate this with the biomarkers discussed in the review article. This can serve as a link between sections 2 and 3, and this will help to understand the discussion part better.

In this review, the authors have utilized a pathophysiologic approach to discuss clinical study results using quantitative biofluid biomarkers in ALS. The review further explores different types of biomarkers for ALS, given ALS’s complexity and the role of different classes related to neuroinflammation, proteinopathy, neurodegeneration, synaptopathy, oxidative stress, aberrant RNA processing, muscle changes, etc. This section covers a discussion of biomarkers related to different factors in ALS but lacks in terms of figure representation. Adding a figure is strongly recommended here for better reach to readers.

The authors should add future directions in the conclusion section for better strategies for biomarker identification. 

The reference list is comprehensive and includes relevant publications. The authors must include the most recent publications (2024) in this article and discuss the most recent findings here.

Author Response

In section 2, the authors discussed the pathophysiology of ALS (familial cases and sporadic cases) and identified several pathogenic mutations that have revealed various mechanisms and genes involved, such as C9orf72, SOD1, TARDBP, FUS, etc. However, the authors should provide a figure on the pathophysiology of ALS. This should cover the molecular pathways and correlate this with the biomarkers discussed in the review article. This can serve as a link between sections 2 and 3, and this will help to understand the discussion part better.

Response: Thank you for pointing this out. We have added 2 figures (figures 1 and 3) to summarize the pathophysiology and the candidate biomarkers, respectively. 

In this review, the authors have utilized a pathophysiologic approach to discuss clinical study results using quantitative biofluid biomarkers in ALS. The review further explores different types of biomarkers for ALS, given ALS’s complexity and the role of different classes related to neuroinflammation, proteinopathy, neurodegeneration, synaptopathy, oxidative stress, aberrant RNA processing, muscle changes, etc. This section covers a discussion of biomarkers related to different factors in ALS but lacks in terms of figure representation. Adding a figure is strongly recommended here for better reach to readers.

response: true. We added a figure for RNA processing (figure 2) and hope that the review will flow better this way. 

The authors should add future directions in the conclusion section for better strategies for biomarker identification. 

response: we added recommendations in red on pages 22 and 23.

The reference list is comprehensive and includes relevant publications. The authors must include the most recent publications (2024) in this article and discuss the most recent findings here.

Response: We looked up studies done in 2024 that we might have missed since we started working on the review and found 3. The changes are in red on pages 13 and 16. 

Reviewer 2 Report (New Reviewer)

Comments and Suggestions for Authors

The study reviews biomarkers for Amyotrophic lateral sclerosis (ALS)  ( a neurodegenerative disease).

Authors described limitations and advances in the field related to ALS diagnostics. Study is interesting but requires some amendments.

1.       Abstract does not reflect the content properly. Some pathophysiology should be mentioned. What kind of biomarkers are the most promising? It should be indicated in the Abstract. Authors mentioned ‘ more objective prognostic tools”, what are they? ‘” Several fluid biomarkers “ – what are they?

2.       Introduction section: Authors should avoid vacuous generalisations.  It is not very clearly presented which biomarkers will be in the focus of this review. Line 77/78: “ These biomarker…” – indicate which biomarkers.

3.        The pathophysiology section should include the diagram of the pathological factors in ALS. See this review https://www.nature.com/articles/nrdp201771 for the example of ALS diagram.

4.       The review has no illustrations. Author should consider making a couple of figures which can improve visualisation of information.

5.        I found that miRs are in focus for this review. Author should make a diagram with miR signaling for ALS. 

Comments on the Quality of English Language

Well written. Only minor editing is recommended.

Author Response

  1. Thank you for your feedback. We've reviewed our abstract and made appropriate changes in red.
  2. We've changed the wording of this sentence (in red) as we cannot enlist all the biomarkers in the introduction. 
  3. True. Figure 1 summarizes the pathophysiology of ALS. 
  4. True. We've made three illustrations for the review and hope they will make it flow better. 
  5. True. We added a figure (figure 3) for the miRNA profile in ALS. 

This manuscript is a resubmission of an earlier submission. The following is a list of the peer review reports and author responses from that submission.

Round 1

Reviewer 1 Report

Comments and Suggestions for Authors

Dear Authors

The article is well-written and scientific with a high-quality structure, but because similar works have been done before, especially in recent years, it is necessary to state in the abstract and introduction part the advantages and novelty of the above review and its differences from the previous ones. 

Reviewer 2 Report

Comments and Suggestions for Authors

Dear authors,

You present “A Review of Biomarkers of Amyotrophic Lateral Sclerosis: A Pathophysiologic Approach” which is a topic of great interest to any researcher/specialist interested in this disease. Although you provide detailed information regarding the existing literature on ALS biomarkers, several things need to be corrected before considering an acceptance for publication.

1)    Abstract: “Additionally, ALS is genetically heterogeneous with more than 40 genes implicated in its pathogenesis”. The pathogenesis and heterogeneity of ALS is not necessarily linked to more than 40 genes. In fact, most ALS cases are sporadic and not be linked to a genetic modification. Only FALS can be correlated to more than 40 different genes, but not to all of them. I suggest reviewing the content of the abstract since a substantial part of it presents well-known facts. The abstract should be focused on ALS biomarkers, which is the transcendental question to discuss.

2)    In section 2, titled: Pathophysiology of Amyotrophic Lateral Sclerosis. The only mentioned pathologic mechanism (line 82) is: “In 97% of ALS cases, the cytoplasmic aggregates are of mislocalized, ubiquitinated phosphorylated TAR DNA binding protein 43-kDa (TDP-43), which normally works as a transcription factor in the nucleus”. Confining the pathophysiology of ALS solely to the presence of TDP-43 aggregates is highly incomplete, biased, and lacks foundation. Moreover, the percentage (97%) seems overstated and is not supported either by the provided bibliographic reference [38] or by epidemiological data. Furthermore, the same sentence/idea is excessively repeated in subsequent paragraphs (lines 130-132, 139-141, 152, 552, and others).

3)    Regarding the possible use of TDP-43 as ALS biomarker, a recent article (Irwin et al. (2024) Fluid biomarkers for amyotrophic lateral sclerosis: a review. Mol Neurodegener) provides a more detailed and extended information than yours. Something that obviously makes the present analysis less interesting. Something similar occurs in other sections.

4) You draw conclusions (lines 152-153) before providing supporting data to the reader (lines 155-180). I also suggest reorganizing the text and removing spaces corresponding to lines 149 and 154.

5)    In line 212: “Using LC-MS, researchers were able to detect significantly elevated levels of a specific SOD1 peptide in CSF of presymptomatic and symptomatic SOD1 mutation carriers (n=13 and n=14, respectively) compared to disease controls comprising patients with neurodegenerative diseases (n=30) and healthy controls (n=29)”. You should provide numerical data. 

6)    In line 285, reference [336] is not correlated with previous/posterior numbered references.

7)  Abbreviatures (superoxide, O2 ), subindexes, spaces between paragraphs, text and references (SOD1[108], [ 110,111,112], [ 116], [277, 278], etc.) must be reviewed.

8)    According to section 3:

a)      The biomarker should be neuron-specific or glial-specific” which is not strictly necessary. 

b)      Prognostic biomarkers can predict an outcome, such as the rate of disease progression and severity, irrespective of treatment”. The prognosis of an illness is related to the existence or absence of treatment, and the response of the patients to the treatment. In that sense, the terrible prognosis of ALS is linked to its pathophysiology but also to the lack of an efficacious treatment. If we had any kind of treatment, a good prognostic biomarker (which could also be named predictive biomarker if  used for a single patient) should differentiate between responders and non-responders (which would ascribe a worse prognosis for the latter).

c)      A good diagnostic biomarker needs to be specific for each disease. In that sense, you wrote: “plasma and CSF NfL can differentiate ALS patients from ALS mimics and other motor neuron diseases (MNDs), highlighting its diagnostic value”. The diagnostic use of plasma and CSF NfL is limited to FALS, precisely because their levels can be elevated in other neurodegenerative conditions. These conceptual mistakes make necessary a complete revision of the present contribution.

9)        In some sections, you miss or skip citing the studies that evidence the lack of correlation or no significative correlation between the proposed biomarker and its blood or CSF levels. To cite an example, in the tofersen trial [116] (line 324)NfL’s potential as a pharmacodynamic biomarker based on their changes in the treated or placebo arm in VALOR trial”.  In the trial, Tofersen was shown to reduce levels of the light neurofilament during, but clinical benefits did not appear until (some) patients reached the open label extension period. Therefore, in this case, NfL was not a good pharmacodynamic biomarker. Another examples of omissions: pNfH has lower diagnostic sensitivity than NfL when its levels are assessed using ELISA techniques; consequently, a clear discussion on whether it is most convenient to use pNfH or NfL as biomarker is missing. These omissions/mistakes also affect to other biomarkers, making  necessary to revise the entire review.  

10)  In general, your provide a literal exposition of the results from the referenced studies (sometimes omitting contradictory results). For your review to become relevant and appealing to readers, as experts in the field, you should compare and comment the significance of the studies/results presented. Along with the lack of foundation in the Pathophysiology section and my comments in point 9, this is one of the most significant limitations of the present review.

11)  The information provided in 611-618 should be part of your introduction. It cannot be 1/3 of the conclusions section of a paper titled: Review of Biomarkers of Amyotrophic Lateral Sclerosis.

I hope my comments are useful to you to improve your work.